# Somatic Embryogenesis in Conifers: One Clade to Rule Them All?

**DOI:** 10.3390/plants12142648

**Published:** 2023-07-14

**Authors:** Hugo Pacheco de Freitas Fraga, Paula Eduarda Cardoso Moraes, Leila do Nascimento Vieira, Miguel Pedro Guerra

**Affiliations:** 1Departamento de Botânica, Setor de Ciências Biológicas, Universidade Federal do Paraná, Curitiba 81530-000, Brazil; hugofraga@ufpr.br (H.P.d.F.F.); paula.eduarda@live.com (P.E.C.M.); leilavieira@ufpr.br (L.d.N.V.); 2Graduate Program in Plant Genetic Resources, Laboratory of Plant Developmental Physiology and Genetics, Federal University of Santa Catarina, Florianópolis 88034-000, Brazil; 3Graduate Program in Agricultural and Natural Ecosystems, Federal University of Santa Catarina, Curitibanos Campus, Ulysses Gaboardi Road, Km 3, Curitibanos 89520-000, Brazil

**Keywords:** gymnosperms, Araucariaceae, Cupressaceae, Gnetales, Pinaceae

## Abstract

Somatic embryogenesis (SE) in conifers is usually characterized as a multi-step process starting with the development of proembryogenic cell masses and followed by histodifferentiation, somatic embryo development, maturation, desiccation, and plant regeneration. Our current understanding of conifers’ SE is mainly derived from studies using Pinaceae species as a model. However, the evolutionary relationships between conifers are not clear. Some hypotheses consider conifers as a paraphyletic group and Gnetales as a closely related clade. In this review, we used an integrated approach in order to cover the advances in knowledge on SE in conifers and Gnetales, discussing the state-of-the-art and shedding light on similarities and current bottlenecks. With this approach, we expect to be able to better understand the integration of these clades within current studies on SE. Finally, the points discussed raise an intriguing question: are non-Pinaceae conifers less prone to expressing embryogenic competence and generating somatic embryos as compared to Pinaceae species? The development of fundamental studies focused on this morphogenetic route in the coming years could be the key to finding a higher number of points in common between these species, allowing the success of the SE of one species to positively affect the success of another.

## 1. Introduction

Plant biotechnologies associated with somatic embryogenesis (SE) provide effective tools to both capture genetic gains and conserve endangered species. They allow the production of large amounts of plant material and also have other benefits [1]. Zimmerman [2] defined SE as a process unique to the plant kingdom that produces normal embryos and whole plants from somatic cells in an undifferentiated state. This morphogenetic route involves dedifferentiation and acquisition of embryogenic competence by somatic cells in response to appropriate inductive signals [3].

More specifically, somatic cells induced to redifferentiate into an embryogenic state reenter the cell cycle while dividing and specialize tissues, such as the bipolar root and shoot meristematic regions [4,5]. This specific developmental potential has been acknowledged both as a valuable pathway for plant regeneration from cell culture systems and as a model for studying early regulatory and morphogenetic events in plant embryogenesis [1].

In addition, SE is favored over the methods of vegetative propagation due to the possibilities of scaling up the propagation by using bioreactors [6] and of enabling processes such as cryopreservation in several plant species, making it possible to store the plant material in the long term. Embryogenic cultures are also an attractive target for genetic modification [7].

Following the early reports of in vitro somatic embryo formation in *Daucus carota* cell suspensions by Steward et al. [8] and Reinert [9], SE has been established in a wide range of plant species. However, plants are very diversified and not all of them are able to undergo such drastic biochemical changes and genetic remodeling, which is referred to as SE recalcitrance [10].

The first report of callus induction in gymnosperms was in 1971 for *Welwitschia mirabilis* (Welwitschiaceae; [11]), and only in 1985 was a complete SE protocol for *Picea abies* (Pinaceae) published [12,13]. Since then, many other gymnosperm genera, such as *Abies*, *Larix*, *Pinus*, *Pseudotsuga*, and *Sequoia* [14,15], have been shown to be able to produce somatic embryos, all of them Pinaceae conifers.

Nowadays, the majority of the SE studies reported are on gymnosperms belonging to the family Pinaceae (mostly the genera *Pinus* and *Picea*), which is widely used as a model system in a great number of SE studies [16,17,18,19,20]. This large number of studies suggests that there might be a preference for working with Pinaceae species in light of the great economic importance of this group in silviculture [21,22,23] and/or that non-Pinaceae conifers might be more recalcitrant to SE and need different stimuli.

With regard to the three main hypotheses for conifers’ evolutionary relationships, two of these hypotheses position conifers as a paraphyletic group and include Gnetales as a closely related clade [24,25,26,27,28,29]. If conifers are paraphyletic, this may reflect important differences in their responses during SE and Gnetales cannot be overlooked.

In this review, we used an integrated approach in order to cover the advances in knowledge on SE in conifers (Pinaceae + non-Pinaceae conifers) and Gnetales, discussing the state-of-the-art and shedding light on similarities and current bottlenecks. With our approach, we expect to be able to better understand the integration of these clades within current studies on SE.

## 2. Taxonomic Considerations

According to the most recent classification of extant gymnosperms proposed by Yang et al. [30], there are 8 orders (Cycadales, Ginkgoales, Araucariales, Cupressales, Pinales, Ephedrales, Welwitschiales, and Gnetales), 13 families, and 86 genera. Also, these authors indicated that gnetophytes constitute a monophyletic group that is a sister group to the Pinaceae (gnepine hypothesis), which was reinforced by phylogenomic results based on thousands of single-copy nuclear genes [30]. These results also negate the monophyly of conifers.

Among conifers (Araucariales, Cupressales, and Pinales), the family Pinaceae (Pinales) stands out. It has the largest number of species (272) distributed across 11 genera, representing ~22.6% of all species of gymnosperms [30]. This family, which typically occurs in the Northern Hemisphere, also has the most economically relevant species in the world, with this aspect possibly being one of the most relevant for the robust scientific production focused on the study of SE in the species of this family.

Comparatively, the other conifer families (Araucariaceae, Cephalotaxaceae, Cupressaceae, Podocarpaceae, Sciadopityaceae, Taxaceae) together have 430 species distributed across 61 genera, representing ~36% of all gymnosperm species [30]. These non-Pinaceae conifers naturally occur in the Southern Hemisphere, except for some species of Cupressaceae and Taxaceae [31]. Many of these species have economic relevance, as will be detailed in other sections of the review.

## 3. Zygotic Embryogenesis in Conifers

Gymnosperms are distinguished by their naked seeds originating in exposed ovules [32]. A large number of gymnosperm species have orthodox seeds, which undergo maturation drying and can withstand dehydration at less than 5% moisture. Many of these species belong to the family Pinaceae and naturally occur in regions with well-defined seasons. However, seeds of many non-Pinaceae conifers show very different behavior related to the maintenance of germination viability when exposed to low moisture content. These recalcitrant seeds are common in many conifer families, such as Araucariaceae and Podocarpaceae, and possibly have distinct, specific requirements for their germination.

Conifer embryos arise from a single fertilization event within the ovule, creating a diploid embryo that develops within a haploid female gametophyte [33]. Most conifers are pollinated and fertilized during the same season, but several species (for example, from the genus *Pinus*) have a period of 12–14 months between ovule pollination and fertilization [34]. In addition, there are differences between species regarding the development and structure of the male gametophyte, the types of pollination mechanisms, the development and behavior of the proembryo, and the occurrence and type of polyembryony [34].

Multiple embryos can commonly be found within the early-stage seeds of conifers [33]. This polyembryony continues until determined embryogeny stages when one embryo within the seed becomes dominant through unknown processes and continues to grow and develop while all others are aborted.

The sequence of embryo development in gymnosperms can be classified into three steps: (i) proembryogeny—all stages before suspensor elongation; (ii) early embryogeny—all stages after suspensor elongation and before the establishment of the root meristem; (iii) late embryogeny—intensive histogenesis including the establishment of the root and shoot meristems [34]. The proembryogeny stage initiates with several rounds of nuclear duplication without cytokinesis and enters a free nuclear phase after fertilization, which is an intriguing and almost exclusive characteristic of gymnosperm embryogeny.

Next, the stages of early and late embryogeny occur. In these stages, the primary body plan is established together with the apical–basal symmetry and the radial symmetry, and then the root and shoot apical meristems are delineated and the plant axis is established [34]. In the late embryogeny stage, the cotyledonary primordia arise in a ring around the shoot apical meristem. Here, some striking differences between Pinaceae and non-Pinaceae conifers can be identified, such as the number of cotyledons, which can range from 2 to 24 [35].

Another stage that demonstrates a series of differences between families, genera, and species of conifers is the process of embryonic maturation. The maturation is responsible for (i) synthesizing large amounts of storage products, (ii) inducing water loss, (iii) preventing premature germination, and (iv) establishing a state of dormancy [34]. Issues associated with seed orthodoxy and recalcitrance can significantly alter the biochemical, physiological, and molecular requirements of this stage, which involves water loss at certain levels. For example, conifers with recalcitrant seeds, such as *Araucaria angustifolia*, certainly exhibit different behaviors compared to certain species of *Picea* that have orthodox seeds.

## 4. Somatic Embryogenesis in Conifers

For several species of Pinaceae, SE is well-characterized as multi-step process starting with the development of proembryogenic masses (PEMs) and followed by early somatic embryo formation, maturation, desiccation, and plant regeneration. To successfully progress along this pathway, several chemical and physical stimuli may be employed [7]. Usually, the following five steps are applied: (i) culturing of the primary explant (in the majority of cases, the megagametophytes and isolated immature or mature zygotic embryos) on a culture medium supplemented with plant growth regulators (PGRs), mainly auxin but often also cytokinin, for initiation of embryogenic cultures; (ii) proliferation of embryogenic cultures on a solidified medium or in a liquid medium supplemented with PGRs, similarly as during initiation; (iii) prematuration of somatic embryos in a culture medium in the absence of PGRs, which inhibits proliferation and stimulates cell differentiation; (iv) maturation of somatic embryos in a high-osmotic-pressure culture medium supplemented with abscisic acid (ABA); and (v) plant development in a PGR-free culture medium [7].

*Picea abies* has been considered the model species for SE studies on conifers, and the first successful report was from 1985 [13]. Since this first report, a series of studies have been able to considerably advance understanding of this species’ morphogenetic route, which has served as a subsidy for the development of countless other SE studies on Pinaceae and non-Pinaceae conifers. However, based on the already-mentioned phylogenetic issues and the particularities of non-Pinaceae conifers associated with their habitat, such as nutritional requirements, etc., the well-established SE model for *P. abies* may not sufficiently represent other conifer species.

Based on this notion, in the following sections, we discuss the SE reports for Ephedrales, Gnetales, Welwitschiales, and non-Pinaceae conifers (Araucariales and Cupressales), aiming to compare SE step-by-step in each of these clades. The criteria used to choose the studies consulted for this review were as follows: (i) articles directly related to the study of somatic embryogenesis in non-Pinaceae conifers, excluding those that only used this technique for another purpose; (ii) articles published in indexed journals up to the year 2022; (iii) articles written in English. The Google Scholar platform was used to search for the cited articles.

## 5. Somatic Embryogenesis in Ephedrales, Gnetales, and Welwitschiales

One of the pioneer studies with gymnosperm SE was performed by Button et al. [11] with *Welwitschia mirabilis* (Welwitchiaceae, Welwitschiales; Table 1). *Welwitschia mirabilis* is a keystone endemic plant from the Namib Desert in Africa and belongs to a monotypic genus (Bombi et al. [36] 2021). This species demands 3 to 20 years to reach maturity and is extremely recalcitrant to vegetative propagation methods [37]. There are few reports on *W. mirabilis* SE, mainly because of the rare availability of experimental material and its slow growth behavior [38]. Bornman [39] successfully performed callus induction from the hypocotyl–root axis of germinating embryos of *W. mirabilis* using Schenk and Hildebrandt (SH) [40] culture medium supplemented with 10 mg L^−1^ 1-naphthaleneacetic acid (NAA). However, somatic embryos could not be obtained.

Somatic embryogenesis studies have also undertaken with species from the genus *Ephedra: Ephedra foliata* and *Ephedra gerardiana* of the order Ephedrales. These species are among the few gymnosperms adapted to arid habitats and they are distributed mainly in semi-arid environments across the Palearctic and Nearctic; nevertheless some species can be found in neotropical countries as well [41,42].

A complete protocol was obtained for SE induction and plant regeneration for *E. foliata* [43]. These authors used zygotic embryos as explants, which were inoculated in MS culture medium supplemented with 2,4-dichlorophenoxyacetic acid (2,4-D) plus kinetin (Kin); the best results obtained were at the lower concentrations tested (2 μM 2,4-D and 2 μM Kin), reaching 60% somatic embryo maturation. It was also found that 6-benzylaminopurine (BAP) plus 2,4-D showed decreased somatic embryo development. The combination of NAA and Kin resulted in root formation and callus induction but with the complete absence of somatic embryo induction.

Somatic embryos of *E. gerardiana* were obtained from internodal segment cultures [44]. These authors found that lower concentrations of thidiazuron (TDZ; 0.5–1.0 µM) in MS culture medium resulted in callus induction in all the cultures initiated after two weeks, and 10–30% of the embryogenic cultures yielded embryoid-like structures. Higher TDZ concentrations (10–20 µM) resulted in embryogenic culture induction and shoot bud formation in 100% of the cultures. These authors showed that the embryoid-like structures obtained from embryogenic cultures turned into somatic embryos two weeks after transfer from a basal medium (PGR-free) and germinated into young plantlets with the same culture medium composition. This indicates that lower TDZ concentrations can induce embryoid-like structures, and higher concentrations of TDZ could be used for indirect organogenesis induction in this species.

The order Gnetales includes one genus: *Gnetum. Gnetum ula* is a large woody climber native to India and a remarkable medicinal plant used to cure a variety of illnesses [45]. The first and only SE report on *G. ula* by Augustine and D’Souza [46] utilized immature zygotic embryos as explants and presented 86% embryogenic cultures induction in MS culture medium supplemented with BAP (5.0 mg/L). The 2,4-D supplementation resulted in non-embryogenic callus formation.

Thus, reports of SE protocols for these three orders are incipient and scarce, and it is certainly necessary to develop further studies, which would allow a better understanding of this morphogenetic route in related species.

## 6. Somatic Embryogenesis in Araucariales

*Araucaria angustifolia* (Araucariaceae, Araucariales) is the most abundant species with SE studies among non-Pinaceae gymnosperms (Table 1). It is a culturally, economically, and ecologically important species in Brazil [47], mainly due to the production of edible seeds and high-quality wood [48,49]. This species is commonly found in south and southeastern Brazil and northwestern Argentina and Paraguay [48], but the remaining *A. angustifolia* forests are under severe pressure from exploration [50]. In this context, much effort has been directed towards developing an effective SE protocol for *A. angustifolia* and expanding the methods available for its mass propagation and conservation.

Astarita and Guerra [51] primarily introduced the application of tissue culture tools with this species. For *A. angustifolia*, the SE model includes two cycles: A, which includes induction and proliferation steps; and B, which covers the maturation phase.

At the beginning of cycle A, the apical portion of the zygotic embryos induces embryogenic cultures in the absence of PGRs or with both auxin and cytokinin supplementation [51,52,53]. Embryogenic cells and suspensor cells (SC) are organized in PEMs [54,55,56]. In the proliferation phase, the PEM I, PEM II, and PEM III stages form the embryogenic culture [57].

The prematuration step triggers early somatic embryo polarization and individualization from PEM III in *A. angustifolia*. This process is usually triggered by PGR removal from the culture medium and its replenishment with maltose and PEG [54,58,59]. Early somatic embryos appear when compact clusters of embryogenic cells grow from PEM III with two regions: the dense globular embryonal mass in the apex and the suspensor in the basal part [60].

Cycle B begins after prematuration, starting with the maturation phase. Here, early somatic embryos rarely turn into late somatic embryos, which can be achieved when the early embryos are capable of responding to the new specific signals with osmotic (i.e., higher levels of gelling agents and PEG supplementation) and hormonal adjustments (mainly ABA supplementation) during the maturation step [54,58]. The early somatic embryo development indicates the beginning of structural differentiation, with protoderm formation around the early somatic embryo followed by meristem determination (root and shoot apical meristems). Afterward, the formation of plantlets may be achieved through the conversion of the somatic embryos [61].

*A. angustifolia* requires complex conditions for SE in comparison to the Pinaceae model systems. This indicates possible biochemical or genetic prerequisites that are not met with those SE model systems. With the large number of biochemical, genetic, and histological studies through the years with these species, some of those prerequisites have been determined [51,52,54,57,58,59,62,63,64,65,66,67,68,69,70].

During the transition from the multiplication to the maturation phase, somatic embryos exhibit rising levels of *ARGONAUTE (AaAGO)*, *CUP-SHAPED COTYLEDON1* (*AaCUC)*, *WUSCHEL-related WOX* (*AaWOX)*, *S-LOCUS LECTIN PROTEIN KINASE* (*AaLecK)*, and *VICILIN* 7S (*AaVIC).* In the proliferation phase, expression of *AaAGO* and *SCARECROW-like* (*AaSCR)* was more common but not *REVERSIBLE GLYCOSYLATED POLYPEPTIDE* (*AaRGP)* or *AaLEC* due to the effect of the presence/absence of both auxin and cytokinin [67,68]. *AaSERK1* could be found as preferentially expressed in embryogenic cell cultures [64]. Furthermore, in situ hybridization results indicated that *AaSERK1* transcription starts to accumulate in groups of cells at the embryogenic culture borders and then becomes limited to the proper developing embryo [64].

Studies involving SE-related gene expression in Pinaceae species have also been reported. Vestman et al. [71] reported that a member of the *ARGONAUTE* family from white spruce is required for proper shoot and root meristem differentiation during embryo development. Important genes for normal embryo patterning, such as putative homologs of *SERK* and *WOX2*, have also been identified as being expressed during both the proliferation and differentiation of early embryos and late embryo development [71]. Rupps et al. [72], investigating SE in *Larix decidua*, also reported transcript accumulation for *LdLEC1* and *LdWOX2* during early embryogenesis, whereas *LdSERK* revealed increased expression during later development. In this context, the expression control of the mentioned genes seems to be conserved during the SE of Pinaceae and non-Pinaceae species.

In addition to SE-specific genes, biochemical markers are important to understanding the regulatory mechanisms of SE. For example, in *A. angustifolia*, a high glutathione (GSH)/glutathione disulfide (GSSG) ratio demand at the beginning of embryogenesis and the presence of nitric oxide (NO) in embryogenic cells may be responsible for maintaining the polarization of proembryos [66]. The manipulation of GSH/GSSG levels and NO emissions have a clear relationship with the viability of early somatic embryos in *A. angustifolia* [59]. These authors reported that low GSH concentrations (0.01 to 0.1 mM) resulted in a scavenger role in the culture medium; nevertheless, they improved the number of early SEs formed in the cell suspension culture medium within a few days of inoculation. In contrast, the development of early precotyledonary embryos was induced by high GSH concentrations (5 mM). The function of NO in those embryo developments was suggested by NO being emitted from their apexes. These findings highlight that the modification of the culture redox state might be a reliable strategy for the development of more efficient embryogenic systems for this species in particular [59].

The effects of PGR supplementation during the SE of this species on global DNA methylation (GDM) and the proteomic profiles of *A. angustifolia* have also been reported [70]. The results showed that the EC induced on PGR-free culture medium had increased GDM compared to the EC induced on PGR-supplemented medium over this period of time, as well as many subcultures, which can be associated with decreased genomic stability and changes in gene expression [70].

SE in *Podocarpus lambertii* (Podocarpaceae), also a member of Araucariales, has been reported [73,74]. This is the only other species from the order Araucariales with a developed SE protocol. This species occurs naturally in the Atlantic Forest biome (Brazil), one of the 25 biodiversity hotspots in the world [75,76]. It is a dioecious species that depends on insects for pollination. Its distribution in restricted areas and the selective economy that always exploits the best specimens with no conservation nor renovation of their communities resulted in this species being listed as “near threatened” and as having a declining population by the IUCN in 2012 [77].

Somatic embryogenesis in *P. lambertii* is correlated with what is usually performed with *A. angustifolia* since they are taxonomically close conifers. Fraga et al. [73] reported an efficient SE protocol for this species with homogeneous somatic embryo formation, especially in the ABA-supplemented treatment of maturation. These authors also showed that GSH supplementation to the maturation culture medium improved the number and quality of somatic embryos. Also, the GSH supplementation affected the endogenous hormone levels during the maturation step. This study opened the possibility of comparative studies of SE in these closely related species, which may result in improvements in maturation and conversion of somatic embryos in *A. angustifolia*.

## 7. Somatic Embryogenesis in Cupressales

Among the studies with non-Pinaceae conifers, more than half concern the order Cupressales (Table 1). From this order, the most representative species is the Japanese cedar (*Cryptomeria japonica*), a monotypic genus with limited distribution in temperate humid regions [78]. Japanese cedar is considered the most important species in Japan, representing 44% of Japan’s forest area [79]. Following *C. japonica*, the genera *Juniperus* and *Taxus* also have some representativity, as shown by the number of SE reports on Cupressales.

*Cryptomeria japonica* (Cupressaceae) has a record of exploration in Japan over the last 1000 years. Today, *C. japonica* is still one of the most valuable afforestation tree species in Japan [80], covering 44% of Japanese artificial forests. In addition, it is a famous object of genetic engineering due to the Japanese cedar pollinosis disease, a result of pollen dispersion from *C. japonica* forests [81]. The high efficiency of genetic transformation and plant regeneration of embryogenic cells from conifers has made them a proper target for genetic transformation [82]. Hence, since the early 2000s, SE and plant regeneration in *C. japonica* have been reported in numerous studies [79,83,84,85,86,87,88].

In this species, SE can only be obtained from immature zygotic embryos [84,86]. Igasaki et al. [84] have also shown that 80% of somatic embryos can germinate and synchronously sprout cotyledons, hypocotyls, and roots when the induced culture medium is supplemented with phytosulfokine (PSK), a small, sulfated plant growth factor peptide [89]. However, Maruyama et al. [86] achieved greater efficiency in germination by using polyethylene glycol (PEG) 6000.

Tanigushi et al. [87] adapted an SE protocol to *C. japonica* from previous reports [83,85]. Up to 80% induction rates from embryogenic cells could be achieved both in open-pollinated seeds of 20 clones of *C. japonica* plus trees and in immature artificially crossed seeds [90,91,92]. The variations in the numbers of somatic embryos in those reports could indicate that somatic embryo formation capacity in *C. japonica* is highly genotype-dependent [80].

Further species from the order Cupressales with extensive SE research belong to the genus *Juniperus*. Species from the genus *Juniperus* (Cupressaceae) grow in several habitats, and they are hardy and adaptable to stress conditions; therefore, they are valuable for long-term rehabilitation purposes. The natural habitat of several *Juniperus* species has been greatly reduced due to overexploitation for timber and fuel wood and clearance for cultivation [93], and several studies on SE in this genus have been performed due to the fact that the species are highly endangered. The majority of the species from the genus *Juniperus* are listed as being of “Least Concern” in the IUCN Red List, and all the species listed in the SE protocols published are also classified in this category. *Juniperus* trees have a great variety of uses, from the wood, which has high value for construction, to the fruits, which have medicinal characteristics that make them suitable for curing headaches and skin diseases [93].

The first report of embryogenic culture induction in a species of the genus *Juniperus* was for *J. oxycedrus* [94]. These authors achieved embryogenic culture induction from leaf explants of adult trees cultured in the presence of 2,4-D or picloram, with 6 to 18% of cultures developing somatic embryos [94].

In *J. excelsa*, it has been shown that MS full-strength culture medium inhibited active callus production, but lower levels of ammonium nitrate (NH_4_NO_3_) or total nitrogen minimized this response [95]. Glutamine supplementation in MS culture medium with reduced nitrogen concentration also increased callus proliferation.

Rapid proliferation and development of somatic embryos were also observed in *J. communis* in PGR-free culture medium with lower nitrogen and calcium levels [93]. Moreover, high initiation frequency for embryogenic cell lines could be obtained from intact megagametophytes at the time of intensive cleavage polyembryogeny. These authors also showed that ABA promoted maturation of early embryos (over 40%), and the germinating embryos retained embryogenic potential in the basal part, leading to the development of new embryogenic tissues.

Based on the study developed for *J. communis*, Belaineh et al. [96] reported a protocol for embryogenic culture induction and maturation in *J. procera*. These authors observed recalcitrance towards embryogenic culture maturation; however, supplementation with ABA slightly stimulated maturation in most of the cultures evaluated [96].

Somatic embryogenesis protocols were also described for four species from the genus *Taxus*: *Taxus brevifolia*, *Taxus cuspidata, Taxus baccata*, and *Taxus wallichiana*. The genus *Taxus* is a natural source of powerful bioproducts, such as paclitaxel, a potent anticancer drug; toxoids, important in the development of antibodies; and flavonoids, which assist in regulating cellular activity and fight free radicals that cause oxidative stress in the body [97]. *Taxus* trees are typically slow-growing and long-lived; as a result, these species have been over-used in past centuries, and this fact, combined with other activities, such as illegal cutting, has contributed to the endangerment of the genus [98].

Considering the obvious importance of this species, several attempts have been made to generate SE protocols. Almost 30 years ago, Ewald et al. [99] aimed to establish an SE protocol for three *Taxus* species (*T. baccata, T. brevifolia*, and *T cuspidata*). These authors evaluated several concentrations/combinations of PGRs in the SPE culture medium [100]. The combination of kinetin, BAP, and 2,4-D showed the best results for embryogenic culture formation for the *Taxus* species, and the obtained embryogenic cultures were able to form proembryonary structures and subcultures after transfer to a culture medium supplemented with kinetin and indole-3-butyric acid (IBA).

The last species from this genus with a published SE protocol is *T. wallichiana* [101]. The authors described an efficient procedure for regeneration of *T. wallichiana* plants from embryogenic cultures derived from zygotic embryos using half-strength WPM culture medium supplemented with BAP and NAA. These authors achieved 10% conversion of the mature somatic embryos into complete plantlets on WPM basal medium supplemented at half strength with activated charcoal, obtaining fully formed plantlets seven to eight months after initiation of culturing. These authors also analyzed the content of paclitaxel and related taxanes and revealed that the accumulation of paclitaxel was higher in the embryogenic cultures than in the non-embryogenic callus.

Several other species of Cupressales also have SE protocols described, despite reports being more specific and scarce. Ahn et al. [102] described an SE protocol for *Chamaecyparis thyoides* without plant growth regulator supplementation. In this protocol, the authors induced embryogenic cultures from megagametophytes containing the precotyledonary zygotic embryos on the modified half-strength embryo maturation culture medium developed by Maruyama et al. [83]. Mature somatic embryos were obtained on half-strength semisolid EM maturation medium supplemented with 50 g L^−1^ maltose, 100 g L^−1^ PEG 4000, and ABA concentrations up to 100 µM [102].

Hu et al. [103] proposed an SE protocol for *Cunninghamia lanceolata*. These authors obtained 13.86% embryogenic culture induction on DCR medium supplemented with 1.5 mg L^−1^ 2,4-D and 0.3 mg L^−1^ kinetin. They also investigated the maturation process and obtained the best results with supplementation with 50 μM ABA and 100 g L^−1^ PEG 6000 [103]. A very efficient protocol has been reported for another Cupressales species, *Cupressus sempervirens* [104]. Despite the strong genotype-dependence observed for this species, the authors were successful in inducing embryogenic cultures in a DCR culture medium supplemented with 10 µM 2,4-D. Maturation occurred in a PGR-free DCR culture medium supplemented with 75 g L^−1^ PEG 4000, and somatic embryos were converted into seedlings in half-strength LP medium supplemented with 2 g L^−1^ activated charcoal.

Liu et al. [105] proposed an SE protocol for *Sequoia sempervirens* using needles from 30-day-old grown seedlings, an unusual type of explant for SE induction in conifers that has the potential to maintain the genotypic fidelity of the donor plant. In this protocol, direct SE was induced on SH culture medium with a combination of BAP (0.5 mg/L) and IBA (1.0 mg/L).

SE has been frequently applied as a potential tool for the in vitro conservation of critically endangered species. Ahn et al. [106] reported a successful protocol for SE induction in *Thuja koraiensis* using megagametophytes with zygotic embryos at precotyledonary and late embryogeny stages. The highest SE induction rate occurred on a culture medium supplemented with 2.2 μM BAP and 4.5 μM 2,4-D, with somatic embryo maturation implemented on a culture medium containing 100 μM ABA [106].

An SE protocol has been reported for another highly endangered species, *Torreya taxifolia* [107]. The authors used a complex PGR combination (0.5 mM 2,4-D, 0.2 mM BAP, 0.2 mM kinetin, 0.1 μM brassinolide, 3.8 μM ABA) and obtained 60 to 100% induction. Somatic embryo maturation occurred on a culture medium supplemented with 37.8 μM ABA and 0.1 μM brassinolide.

**Table 1 plants-12-02648-t001:** Somatic embryogenesis (SE) protocols published for non-Pinaceae conifers up to 2022.

Order	Species	Explant Used	Formulation of Culture Medium	Steps of the Protocol	Reference
Araucariales	*Araucaria angustifolia*	Zygotic embryos	LP salts + vitamins + 0.5 g L^−1^ nicotinic acid + 45 µM 2,4-D + 11.0 µM KIN + 11.0 µM BAP + 500 mg L^−1^ casein + 1% PEG 8000	Embryogenic culture induction	Astarita et al. [51]
		Zygotic embryos	LP salts + 6.8 µM 2,4-D + 2.3 µM KIN + 2.2 µM BAP + 1% PEG 8000 + 0.9 µM sucrose + 38 µM ABA	Embryogenic culture induction	Astarita et al. [62]
		Zygotic embryos	BM salts + vitamins + 5 µM 2,4-D + 2 µM BAP + 2 µM KIN BM salts + vitamins + 6% and 9% PEG 3350 + 6% and 9% BAP + 1 µM KIN	Embryogenic culture induction Precotyledonary somatic embryo development	dos Santos et al. [52]
		Zygotic embryos	LP salts + vitamins + 5 µM 2,4-D + 2 µM BAP + 2 µM KIN LP salts + vitamins + 5.0 µM ABA + 1% PEG 4000	Embryogenic culture induction Precotyledonary somatic embryo development	Silveira et al. [65]
		Zygotic embryos	BM salts + vitamins + 5 µM 2,4-D + 2 µM BAP + 2 µM KIN BM salts + vitamins + 5 µM 2,4-D + 2 µM BAP + 2 µM KIN + 9% PEG + 9% maltose	Embryogenic culture induction Early precotyledonary somatic embryo development	Steiner et al. [58]
		Zygotic embryos	BM salts + vitamins + 2.0 µM 2,4-D + 0.5 µM BAP + 0.5 µM KIN + 0.5 g L^−1^ casein + 1.0 µM Spd + 10. µM Spm	Induction of stage-three proembryogenic masses (PEM III)	Silveira et al. [66]
		Zygotic embryos	BM salts + vitamins + 0.5 g L^−1^ casein + 1 µM Put	Embryogenic culture induction	Steiner et al. [62]
		Zygotic embryos	BM salts + 5 µM 2,4-D + 2 µM BAP + 2 µM KIN	Embryogenic culture induction	dos Santos et al. [54]
		Zygotic embryos	BM salts + 5.5 µM myo-inositol + 0.008 µM nicotinic acid + 0.005 µM pyridoxyn-HCl + 0.05 µM glycine + 0.006 µM thiamine-HCl + 6.8 mM L-glutamine + 0.05% casein hydrolysate + 3% sucrose	Embryogenic culture induction	Maurer et al. [108]
		Zygotic embryos	MSG salts + 0.005 g L^−1^ nicotinic acid + 0.005 g L^−1^ pyridoxine-HCl + 0.001 g L^−1^ thiamine + 1.46 g L^−1^ L-glutamine + 30 g L^−1^ sucrose + 3 g L^−1^ gelrite MSG salts + 5 µM 2,4-D + 2 µM BAP + 2 µM KIN MSG salts + vitamins + 1.46 g L^−1^ L-glutamine + 3 g L^−1^ gelrite + 120 µM ABA + 9% maltose + 7% PEG 4000 + 3% sucrose + 0.15% AC	Embryogenic culture induction Embryogenic proliferation Proembryo formation	Schlögl et al. [68]
		Zygotic embryos	BM salts + vitamins + 0.5 g L^−1^ casein hydrolysate + 1 g L^−1^ myo-inositol + 0.1 g L^−1^ L-glutamine + 3 g L^−1^ sucrose + 5 µM 2,4-D + 2 µM BAP + 2 µM KIN MSG salts + vitamins + 1.46 g L^−1^ L-glutamine + 90 g L^−1^ maltose + 70 g L^−1^ PEG 3350 + 1 µM GSH + 100 µM GSSG + 30 g L^−1^ sucrose	Embryogenic culture induction Proembryo formation	Vieira et al. [59]
		Zygotic embryos	LP salts + 10 g L^−1^ sucrose + 0.45 g L^−1^ L-glutamine BM salts + 30 g L^−1^ sucrose + 0.5 g L^−1^ casein hydrolysate + 1 g L^−1^ myo-inositol + 1.0 g L^−1^ L-glutamine DKM salts + 0.5 g L^−1^ casein hydrolysate + 0.1 g L^−1^ myo-inositol + 30 g L^−1^ sucrose + 30 µM fluridone DKM salts + 0.5 g L^−1^ casein hydrolysate + 0.1 g L^−1^ myo-inositol + 90 g L^−1^ maltose + 70 g L^−1^ PEG 3350	Embryogenic culture induction Embryogenic culture proliferation Pre-maturation Proembryo formation	Farias-Soares et al. [109]
		Zygotic embryos	MSG salts + 1.46 µM L-glutamine + 30 g L^−1^ sucrose + 3 g L^−1^ gelrite + 1.0 µM 2,4-D + 0.5 µM BAP MSG salts + 1.46 g L^−1^ L-glutamine + 30 g L^−1^ sucrose + 3 g L^−1^ AC + 70 g L^−1^ maltose + 90 g L^−1^ PEG 4000	Embryogenic culture induction Cotyledonary somatic embryo development	Jo et al. [110]
		Zygotic embryos	MSG salts + 120 µM ABA + 90.0 g L^−1^ PEG 4000 + 3.0 g L^−1^ maltose	Early precotyledonary somatic embryo development	Elbl et al. [111]
		Zygotic embryos	BM salts + 1.0 g L^−1^ L-glutamine + 1.0 g L^−1^ myo-inositol + 0.5 g L^−1^ casein hydrolysate + 30 g L^−1^ sucrose BM salts + 1.0 g L^−1^ L-glutamine + 1.0 g L^−1^ myo-inositol + 0.5 g L^−1^ casein hydrolysate + 50 g L^−1^ maltose + 100 g L^−1^ PEG 4000 + 100 μM ABA	Embryogenic culture induction Precotyledonary somatic embryo development	Fraga et al. [69]
		Zygotic embryos	DKM salts + 0.5 g L^−1^ casein hydrolysate + 0.1 g L^−1^ myo-inositol + 30.0 g L^−1^ sucrose + 30 μM fluridone DKM salts + 0.5 g L^−1^ casein hydrolysate + 0.1 g L^−1^ myo-inositol + 90.0 g L^−1^ maltose or 90.0 g L^−1^ lactose + 70.0 g L^−1^ PEG	Embryogenic culture induction Proembryo formation	Steiner et al. [57]
		Zygotic embryos	MSG salts + 1.46 g L^−1^ L-glutamine + 3% sucrose MSG salts + 1.46 g L^−1^ L-glutamine + 3% sucrose + 120 µM ABA + 7% maltose + 9% PEG 4000	Embryogenic culture induction Early somatic embryo development	Navarro et al. [112]
	*Podocarpus lamberti*	Zygotic embryos	MSG salts + BM vitamins + 1.46 g L^−1^ L-glutamine + 0.1 g L^−1^ myo-inositol + 30 g L^−1^ sucrose MSG salts + BM vitamins + 1.46 g L^−1^ L-glutamine + 0.1 g L^−1^ myo-inositol + 50 g L^−1^ maltose + 100 g L^−1^ PEG 4000 + 2 g L^−1^ AC + 75 µM ABA + GSH 500 µM	Embryogenic culture induction Cotyledonary somatic embryo development	Fraga et al. [73]
		Zygotic embryos	MSG salts + BM vitamins + 30 g L^−1^ sucrose MSG salts + BM vitamins + 50 g L^−1^ maltose + 100 g L^−1^ PEG 4000 + 2 g L^−1^ AC + 75 µM ABA	Embryogenic culture induction Cotyledonary somatic embryo development	Guerra et al. [74]
Welwitschiales	*Welwitschia mirabilis*	Zygotic embryos	MS salts + 0.5 g L^−1^ glycine + 5.0 g L^−1^ inositol + 0.003 g L^−1^ calcium pantothenate + 0.03 g L^−1^ thiamine hydrolysate + 40 g L^−1^ sucrose + 100/50 mg L^−1^ NAA	Callus induction	Button et al. [11]
		Zygotic embryos	SH salts + 0.3 mg L^−1^ NAA	Callus induction	Bornman [39]
		Leaf	MS salts + 9.0/22.5/45.0 µM 2,4-D + 2.2/4.4 µM BAP + 4.6 µM KIN + 5.4 µM NAA	Embryogenic culture induction	Misra et al. [38]
Gnetales	*Gnetum ula*	Megagametophyte + immature zygotic embryos	MS salts + 20 g L^−1^ sucrose + 5 mg L^−1^ BAP MS/2 salts (PGR-free)	Embryogenic culture induction Immature somatic embryo formation	Augustine and D’Souza [46]
Ephedrales	*Ephedra foliata*	Zygotic embryos	MS salts + 2 µM 2,4-D + 10 µM KIN MS salts PGR-free	Embryogenic culture induction and somatic embryo maturation Somatic plantlet formation	Dhiman et al. [43]
	*Ephedra gerardiana*	Internodal segments	MS salts + 1.0 µM TDZ MS salts PGR-free	Embryogenic culture induction and somatic embryo maturation Somatic plantlet formation	Sharma et al. [44]
Cuppressales	*Cupressus sempervirens*	Zygotic embryos	DCR salts + 0.5 g L^−1^ casein hydrolysate + 0.2 g L^−1^ myo-inositol + 0.1 g L^−1^ L-glutamine + sucrose (30 g/L) + 10 µM 2,4-D DCR salts + 1 µM ABA + 0.02% AC	Embryogenic culture induction Somatic embryo formation	Lambardi et al. [113]
		Zygotic embryos	MS salts + 15 g L^−1^ fructose + 15 g L^−1^ glucose + 4 g L^−1^ AC + 10 mL l^−1^ coconut water MS + 15 g L^−1^ fructose + 15 g L^−1^ glucose + 4 g L^−1^ AC + 1 g L^−1^ BSA	Embryogenic culture induction Proembryo formation	Sallandrouze et al. [114]
		Megagametophyte + immature zygotic embryos	DCR salts + 0.5 g L^−1^ casein hydrolysate + 0.2 g L^−1^ myo-inositol + 0.1 g L^−1^ L-glutamine + 30 g L^−1^ sucrose + 10 µM 2,4-D DCR salts + 75 g L^−1^ PEG 4000	Embryogenic culture induction Somatic embryo development	Barberini et al. [104]
		Megagametophyte + immature zygotic embryos	DCR salts + 0.5 g L^−1^ casein hydrolysate + 0.2 g L^−1^ myo-inositol + 0.1 g L^−1^ L-glutamine + sucrose (30 g/L) + 10 µM 2,4-D DCR salts + 75 g L^−1^ PEG 4000	Embryogenic culture induction Somatic embryo development	Lambardi et al. [115]
	*Cryptomeria japonica*	Zygotic embryos	CDm salts + 0.8 g L^−1^ ammonium nitrate + 1 μM 2,4-D + 6.0 g L^−1^ L-glutamine CDm salts + ABA (0, 0.1, 1, and 10 µM) + BAP (0.1, 1, and 10 µM)	Embryogenic culture induction Proembryo development	Ogita et al. [116]
		Zygotic embryos	SM1 salts + 10 g L^−1^ sucrose + 10 μM 2,4-D + 3 μM BAP + 0.5 g L^−1^ L-glutamine LPM salts + 10 g L^−1^ sucrose + 10 μM 2,4-D + 3 μM BAP + 0.5 g L^−1^ L-glutamine SM3 salts + PEG 400 + ABA LP salts + 0.2 g L^−1^ sucrose + 5 g L^−1^ AC	Embryogenic culture induction Somatic embryo development Somatic plantlet formation	Maruyama et al. [83]
		Zygotic embryos	MSG salts + 0.01% myo-inositol + 0.15% L-glutamine + 3.2 μM 2,4-D + 1.8 μM BAP + 3% sucrose + 0.4% gellan gum EMM salts + 0.2% AC + 0.3% gellan gum EMM salts + 0.2% AC + 3/5-strength EMM vitamins + 400 mg L^−1^ L-glutamine + 260 mg L^−1^ arginine + 20 mg L^−1^ proline	Embryogenic culture induction Somatic embryo development Somatic plantlet formation	Igasaki et al. [84]
		Zygotic embryos	MSG salts + 0.01% myo-inositol + 0.15% L-glutamine + 3% sucrose + 3.2 μM 2,4-D + 1.8 μM BAP + 32 nM PSK EMM salts + 5% PEG 4000 + 3% maltose + 100 μM ABA + 32 nM PSK + 0.2% AC + 0.3% gellan gum SGM salts + 0.2% AC + 10 μM GA_3_	Embryogenic culture induction Somatic embryo development Somatic plantlet formation	Igasaki et al. [85]
		Zygotic embryos	mCD salts + 3% sucrose + 4.1 mM L-glutamine + 1 µM 2,4-D mCD salts + 3% sucrose + 16.4 mM L-glutamine +100 µM ABA + 6% maltose	PEM induction PEM maturation	Nakagawa et al. [117]
		Zygotic embryos	MSG salts + 0.01% myo-inositol + 0.15% glutamine + 3.2 µM 2,4-D + 1.8 µM BAP + 3% sucrose + 0.4% gellan gum	Embryogenic culture induction and somatic embryo development	Igasaki et al. [118]
		Megagametophyte	EM salts + 10 g L^−1^ sucrose + 10 μM 2,4-D + 5 μM BAP + 0.5 g L^−1^ casein + 0.5 g L^−1^ L-glutamine EM salts + 30 g L^−1^ maltose + 2 g L^−1^ AC + 100 μM ABA + amino acids + 150 g L^−1^ PEG + 3.3 g L^−1^ gellan gum EM salts + 20 g L^−1^ sucrose + 1.5 g L^−1^ L-glutamine + 10 g L^−1^ agar	Embryogenic culture induction Cotyledonary embryo development Somatic plantlet formation	Maruyama et al. [119]
		Zygotic embryos	MS salts + 30 g L^−1^ sucrose + 3 μM 2,4-D + 1 μM BAP MS salts + 30 g L^−1^ maltose + 175 g L^−1^ PEG 6000 + 2 g L^−1^ AC + 100 μM ABA	Embryogenic culture induction Somatic embryo development	Izuno et al. [88]
		Seed	EM salts + 10 g L^−1^ sucrose + 10 μM 2,4-D + 5 μM BAP + 0.5 g L^−1^ casein + 0.5 g L^−1^ L-glutamine + 3 g L^−1^ gelrite EM salts + 30 g L^−1^ sucrose + 3 mΜ 2,4-D + 1 μM BAP + 1.5 g L^−1^ L-glutamine + 3 g L^−1 −1^ gelrite EM salts + 175 g L^−1^ PEG + 100 µM ABA + 2 g L^−1^ L-glutamine + 1 g L^−1^ asparagine + 0.5 g L^−1^ arginine + 3 g L^−1^ gelrite	Embryogenic culture induction Embryogenic culture proliferation Somatic embryo maturation	Maruyama et al. [79]
	*Chamaecyparis pisifera*	Immature seed	MS salts + 0.5 g L^−1^ L-glutamine + 2,4-D + BAP MS salts + 100 μM ABA + 2 g L^−1^ AC + 150 g L^−1^ PEG 4000 LP salts + 30 g L^−1^ sucrose + 5 g L^−1^ AC + 12.5 g L^−1^ Wako agar	Embryogenic culture induction Somatic embryo maturation Somatic plantlet formation	Maruyama et al. [120]
		Immature seed	MS salts + 0.5 g L^−1^ L-glutamine + 2,4-D + BAP MS salts + 100 μM ABA + 2 g L^−1^ AC + 150 g L^−1^ PEG 4000 LP salts + 30 g L^−1^ sucrose + 5 g L^−1^ AC + 12.5 g L^−1^ Wako agar	Embryogenic culture induction Somatic embryo maturation Somatic plantlet formation	Maruyama et al. [121]
		Immature seed	MS/2 salts + 10 μM 2,4-D + 5 μM BAP + 10 g L^−1^ sucrose MS/2 salts + 1–10 μM 2,4-D + 0.3 μM BAP + 0.5 g L^−1^ L-glutamine + 30 g L^−1^ sucrose EM salts + 50 g L^−1^ maltose + 100 μM ABA + 2 g L^−1^ AC + 150 g L^−1^ PEG 4000 + amino acids EM/2 salts + 2 g L^−1^ AC + 10 g L^−1^ agar	Embryogenic culture induction Embryogenic culture proliferation Somatic embryo maturation Somatic plantlet formation	Hosoi and Maruyama [122]
	*Chamaecyparis obtusa*	Zygotic embryos	MS salts + 10 μM 2,4-D + 30 g L^−1^ sucrose MS salts + vitamins + 100 μM ABA + 150 g L^−1^ PEG 4000 + 30 g L^−1^ maltose MS salts + 2 g L^−1^ AC + 20 g L^−1^ sucrose + 5 g L^−1^ gelrite	Embryogenic culture induction Cotyledonary embryo development Somatic plantlet formation	Konogaya et al. [123]
		Zygotic embryos	EM salts + 0.40 g L^−1^ KCl + 0.5 g L^−1^ casein + 1 g L^−1^ L-glutamine + 10 g L^−1^ sucrose + 10 µM 2,4-D + 5 µM BAP EM salts + 50 g L^−1^ maltose + 100 g L^−1^ PEG 4000 + 2 g L^−1^ AC + 100 µM ABA + EMM amino acids	Embryogenic culture induction Somatic plantlet formation	Maruyama et al. (2005) [124]
	*Chamaecyparis thyoides*	Zygotic embryos	EM salts + 0/4.5/9 µM 2,4-D + 0/2.2/4.4 µM BAP + 10 g L^−1^ sucrose + 0.5 g L^−1^ myo-inositol + 1 g L^−1^ L-glutamine EM salts + 50 g L^−1^ maltose + 100 gL^−1^ PEG 4000 + 0/10/50/100 µM ABA + 0/2 g L^−1^ AC EM salts + 10 g L^−1^ sucrose + 2 g L^−1^ AC	Embryogenic culture induction Somatic embryo development Somatic plantlet formation	Ahn et al. [102]
	*Cunninghamia lanceolata*	Zygotic embryos	DCR salts + g L^−1^ sucrose + 0.5 g L^−1^ casein hydrolysate + 0.4 g L^−1^ L-glutamine + 5 g L^−1^ Phytagel + 1 g L^−1^ AC + 0.5 mg L^−1^ 2,4-D + 0.5 mg L^−1^ KIN DCR salts + 50 μM ABA + 100 g L^−1^ PEG 6000	Embryogenic culture induction Somatic embryo development	Hu et al. [103]
		Zygotic embryos	DCR salts + 1–2 mg L^−1^ 2,4-D + 0.5 mg L^−1^ KIN + 0.5 mg L^−1^ BAP + 1 mg L^−1^ vitamin C + 0.45 g L^−1^ L-glutamine + 20 g L^−1^ maltose + 2.5 g L^−1^ AC + 2.3 g L^−1^ gelrite DCR salts + 2–8 mg L^−1^ ABA + 1.5 mg L^−1^ GA + 0.2 g L^−1^ proline + 0.45 g L^−1^ L-glutamine + 1 mg L^−1^ vitamin C + 0.5 g L^−1^ casein hydrolysate + 25 g L^−1^ maltose + 2 g L^−1^ AC + 0.2 g L^−1^ aspartic acid + 2.8 g L^−1^ gelrite + 100–200 g L^−1^ PEG	Embryogenic culture induction Somatic embryo development	Wang et al. [125]
		Zygotic embryos	DCR salts + 2.0–6.0 mg L^−1^ 2,4-D + 0.5 mg L^−1^ BAP + 500 mg L^−1^ casein hydrolysate + 450 mg L^−1^ L-glutamine + 100 mg L^−1^ myo-inositol + 20 g L^−1^ maltose + 2.1 g L^−1^ gellan gum DCR salts + 3 mg L^−1^ ABA + 1.0 mg L^−1^ GA_3_ + 500 mg L^−1^ casein hydrolysate + 120–200 g L^−1^ PEG 8000 + 30 g L^−1^ maltose	Embryogenic culture induction Somatic embryo development	Zhou et al. [126]
	*Torreya taxifolia*	Zygotic embryos	2207 salts + 0.25% AC + 15 g L^−1^ maltose + 1 g L^−1^ myo-inositol + 110 mg L^−1^ + 2,4-D + 45 mg L^−1^ BAP + 43 mg L^−1^ KIN + 0.048 mg L^−1^ brassinolide + 1 mg L^−1^ ABA + 5 mg L^−1^ biotin + 50 mg L^−1^ folic acid + 250 mg L^−1^ MES + 60.7 mg L^−1^ pyruvic acid + 450 mg L^−1^ L-glutamine 2207 salts + 1% AC + 15 g L^−1^ maltose + 200 mg L^−1^ ABA + 5 mg L^−1^ biotin + 0.1000 μM brassinolide + 50 mg L^−1^ folic acid + 250 mg L^−1^ MES + 60.7 mg L^−1^ pyruvic acid + 0.048 mg L^−1^ brassinolide + 450 mg L^−1^ L-glutamine	Embryogenic culture induction Somatic embryo development	Ma et al. [107]
	*Sequoia sempervirens*	Needles	SH salts + 0/0.05/0.1 g L^−1^ BAP + 0/0.02/0.05 g L^−1^ KT + 0.02/0.05/0.1 g L^−1^ IBA + 0.05% casein hydrolysate SH salts + 0.05 g L^−1^ IBA + 0.01 g L^−1^ NAA + 0.1% AC + 1.5% sucrose + 0.05% casein hydrolysate	Embryogenic culture induction Somatic embryo development	Liu et al. [105]
	*Juniperus communis*	Zygotic embryos	LP salts + 15 mM ammonium nitrate + 30 g L^−1^ sucrose + 9 µM 2,4-D + 4.4 µM BAP + 0.044 g L^−1^ L-glutamine LP salts + 15 mM ammonium nitrate + 30 g L^−1^ sucrose + 32 mg L^−1^ ABA + 0.044 g L^−1^ L-glutamine	Embryogenic culture induction Proembryo development	Belaineh et al. [96]
		Megagametophytes	LP salts (PGR-free) DCR salts + 2.02 mM potassium nitrate + 1.16 mM calcium chloride + 60 µM ABA + 20 or 100 µM GA_4/7_	Embryogenic culture induction Early somatic embryo development	Helmerson et al. [93]
	*Juniperus procera*	Zygotic embryos	LP salts + 15 mM ammonium nitrate + 3.0 g L^−1^ sucrose + 9.0 µM 2,4-D + 4.4 µM BAP + 0.044 g L^−1^ L-glutamine + 1.6 g L^−1^/3.2 g L^−1^/6.4 g L^−1^/12.8 g L^−1^/25.6 g L^−1^ ABA + 7.5% PEG 4000	Non-embryogenic callus	Belaineh et al. [96]
	*Juniperus excelsa*	Shoots	MS salts + ammonium nitrate/potassium nitrate + L-glutamine	Callus induction	Shanjani et al. [95]
	*Thuja koraiensis*	Zygotic embryos	EM salts + SH + Litvay + 2.2 µM BAP + 4.5 µM 2,4-D + 1 g L^−1^ L-glutamine + 0.5 g L^−1^ myo-inositol + 1 g L^−1^ AC + 10 g L^−1^ sucrose IM salts + 1% sucrose	Embryogenic culture induction Somatic plantlet formation	Ahn et al. [106]
	*Taxus cuspidata*	Zygotic embryos	SPE salts + 0.001 g L^−1^ KIN + 0.002 g L^−1^ NAA	Embryogenic culture induction	Ewald et al. [99]
	*Taxus brevifolia*	Zygotic embryos	SPE salts + 0.001 g L^−1^ KIN + 0.002 g L^−1^ NAA	Embryogenic culture induction	Ewald et al. [99]
	*Taxus baccata*	Zygotic embryos	SPE salts + 0.001 g L^−1^ KIN + 0.002 g L^−1^ NAA	Embryogenic culture induction	Ewald et al. [99]
	*Taxus wallichiana*	Zygotic embryos	B5 salts + SH vitamins + 3% sucrose + 0.01 g L^−1^ NAA + 0.005 g L^−1^ BAP WPM salts + SH vitamins + 3% sucrose + 0.02 g L^−1^ NAA + 0.05 g L^−1^ BAP WPM salts + 0.01 g L^−1^ ABA + 1.0% AC	Embryogenic culture induction Early precotyledonary somatic embryo development Somatic plantlet formation	Datta and Jha [101]

MS—Murashige and Skoog [127]; WPM—woody plant medium [128]; B5—Gamborg et al. [129]; BM—Gupta and Pullman [130]; MSG—Becwar et al. [131]; LM—Litvay et al. [132]; DKM—von Arnold and Clapham [133]; EM—embryo maturation medium [83]; LP culture medium [96]; DCR—Gupta and Durzan [134]; SPE—Gupta and Durzan [100]; 2207 [135]; Cd culture medium [136]; EMM/SGM—Smith’s germination medium [137]; 2,4-D—2,4 dichlorophenoxyacetic acid; GA_3_—gibberellic acid; TDZ—thidiazuron; BA—benzyladenine; BAP—6-benzylaminopurine; NAA—1-naphtaleneacetic acid; KIN—kinetin; ABA—abscisic acid; IBA—indole-3-butyric acid; AC—activated charcoal; PGR—plant growth regulator; BSA—buthionine sulfoximine; GSSG—glutathione disulfide; GSH—glutathione reduced; Put—putrescine; Spm—spermine; Spd—spermidine; PEG—polyethylene glycol.

## 8. Conclusions and Future Directions

Some issues encountered in triggering SE in conifers have not yet been solved despite the assiduous research efforts over time. While most of the conditions required for SE in Pinaceae species are known, somatic embryo formation and development in many non-Pinaceae conifers are hampered. Non-Pinaceae conifers’ recalcitrance towards SE in general—for example, the slow-growing nature of species, such as *W. mirabilis*, or the complex conditions required for a complete SE protocol, such as with *A. angustifolia*—may be an obstacle to obtaining satisfactory protocols.

A better understanding of SE in non-Pinaceae conifers should prove fruitful in the future as SE is now one of the most critical methods for the large-scale propagation of elite genotypes and their conservation. This is especially important because most non-Pinaceae conifers have endangered status.

Some level of understanding has indeed been achieved for most non-Pinaceae conifers, but it is also true that more coordinated research on the SE of these species is urgent. Analyzing the main variables tested in the protocols described in this review, it is possible to list some common points and many dissimilarities in SE protocols for non-Pinaceae conifers. Among the main points held in common are: (i) the use of zygotic embryos as the preferable explant; (ii) the use of strong auxins, such as 2,4-D, in the induction step; (iii) the use of ABA and osmotic agents (such as PEG), sometimes combined with higher concentrations of gelling agents, during the maturation stage; and (d) the use of activated carbon in the conversion to somatic seedlings step. Despite these points in common, there are many heterogeneities in the described protocols, such as: (i) the use of different saline formulations, often in protocols for the same species; (ii) the use of various vitamin compositions; (iii) supplementation of the culture medium with different amino acids or polyamines; and (iv) non-standardization in the carbon sources used.

Therefore, we expect to see increased integration of non-Pinaceae species in future SE studies considering the higher phylogenetic proximity that they present. The development of fundamental studies focused on this morphogenetic route in the coming years could be the key to finding a higher number of points in common between the described protocols, allowing the success of the SE of one species to positively affect the success of another.

## Data Availability

No new data were created or analyzed in this study. Data sharing is not applicable to this article.

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
