# Peer review of "Somatic Embryogenesis in Conifers: One Clade to Rule Them All?"

_plants, 2023, doi:10.3390/plants12142648_

Round 1

Reviewer 1 Report

The authors provide a comprehensive overview of the studies dealing with somatic embryogenesis in conifers, including Gnetales, that are placed as sister to Pinaceae according to recent phylogenomic studies. Somatic embryogenesis (SE) in Pinaceae is usually a well-established system, and the authors mentioned it only shortly, providing a list of articles published from 2010 up to the year 2022 in Table 1. This review is focused mainly on studies on somatic embryogenesis in non-Pinaceae species, such as Gnetales, Araucariales, and Cupressales. The list of SE protocols published for these species until 2022 is provided in Table 2. The conclusion, at least for me, is that the implementation of procedures well-established in Pinaceae SE protocols is not fully successful in non-Pinaceae species. Probably different approaches will be necessary, based on the genomic and phytohormonal knowledge.  

- The review is quite well written, although I would highly recommend proofreading or the usage of some grammar/style software to make the text more understandable for readers. More comments are in the Language quality part.

- I would like to draw your attention to Figure 1, which is incorrect. Genera Cupressus, Sequoia, Taxus and Juniperus do not belong to Araucariales. In the figure,  these genera are marked blue ( as Araucaria and Podocarpus) instead of orange. 

- In part 4. SE in Araucariales, paragraph starting on line 188 - you mentioned cycle B, starting after pre-maturation phase. Could you, please, specify what kind of "new specific signals with osmotic and hormonal adjustments during maturation step" you mean?  

- References - please, pay attention to proper citing of published articles. In Table 1, there are mistakes in family names (Vandrakova instead of Vondrakova; Hazubska-Prybyl instead of Hazubska-Przybyl),  wrong year of publication - Krajnaková et al. 1013 instead of 2013. In the text, lines 184 and 191 - there are references to Santos et al 2008, but in the list of references is dos Santos et al.

My main criticism is that the reference list at the end of the manuscript lacks the references cited in Tables 1 and 2. It is necessary to add full citations to the list of references. If it is not possible due to the number of them, I would highly recommend adding information to both tables. There should be full information about the reference (shorten name of the journal, volume, issue, and pages, not only the year of publication) or/and DOI. Otherwise, the articles are unsearchable.

I consider these reproofs as minor. Whenever the manuscript will be corrected/amended I would approve it for publication.

As I have mentioned above, I would recommend proofreading to improve the clarity of the manuscript. I do not want to rewrite the whole manuscript as I am not qualified for this work. At this place, I just want to propose only several improvements.

lines 170-172: I would recommend improving this sentence: ” From this perspective, many efforts were made to develop an efficient SE protocol for A. angustifolia and expand available methods its mass propagation and conservation.” - From this perspective, much effort has been made to develop an effective SE protocol for A. angustifolia and to expand the methods available for its mass propagation and conservation.

lines 182-184: “The pre-maturation step is the trigger to early SE polarization and individualization from PEM III in A. angustifolia. Removing the PGR from the culture medium and its supplementation with maltose, and PEG usually trigger this process”. - The pre-maturation step triggers early somatic embryo polarization and individualization from PEM III in A. angustifolia. This process is usually triggered by the removal of PGR from the culture medium and its replenishment with maltose and PEG. (or The pre-maturation step, when PGRs are replaced with maltose and PEG in the culture medium, induces polarization of early somatic embryos in A. angustifolia and their individualization from PEM III.) 

If I well understood the sentence, I would propose using "early somatic embryos" because the abrev. SE is defined for somatic embryogenesis. 

Lines 213 – 214: I do not understand this sentence: “Additionally, SE-specific genes, biochemical markers are important to understanding the regulatory mechanisms of SE.” Do you mean “In addition to SE-specific genes, biochemical markers are also important for understanding the regulatory mechanisms of SE”?  

-I would like to draw your attention also to some typing errors - in the abstract, line13 - hystodifferentiation instead of histo...; line 96 - Araucariales e Cupressales instead of A. and C.?

Author Response

- The review is quite well written, although I would highly recommend proofreading or the usage of some grammar/style software to make the text more understandable for readers. More comments are in the Language quality part.

R: We accepted the comments and revised the text, as suggested.

- I would like to draw your attention to Figure 1, which is incorrect. Genera Cupressus, Sequoia, Taxus, and Juniperus do not belong to Araucariales. In the figure,  these genera are marked blue ( as Araucaria and Podocarpus) instead of orange.

R: We omitted this figure, as suggested by reviewer 2.

- In part 4. SE in Araucariales, paragraph starting on line 188 - you mentioned cycle B, starting after the pre-maturation phase. Could you, please, specify what kind of "new specific signals with osmotic and hormonal adjustments during maturation step" you mean?

R: We included this information, as requested.

- References - please, pay attention to proper citing of published articles. In Table 1, there are mistakes in family names (Vandrakova instead of Vondrakova; Hazubska-Prybyl instead of Hazubska-Przybyl),  wrong year of publication - Krajnaková et al. 1013 instead of 2013. In the text, lines 184 and 191 - there are references to Santos et al 2008, but in the list of references is dos Santos et al.

R: We omitted Table 1, as suggested by reviewer 2. Also, we revised all references and corrected them.

- My main criticism is that the reference list at the end of the manuscript lacks the references cited in Tables 1 and 2. It is necessary to add full citations to the list of references. If it is not possible due to the number of them, I would highly recommend adding information to both tables. There should be full information about the reference (shorten the name of the journal, volume, issue, and pages, not only the year of publication) or/and DOI. Otherwise, the articles are unsearchable.

R: We screened all tables and reference lists to include as much information as possible to improve the traceability of cited articles and chapters. In Table 1 we added the DOI in all the references with the link.

- lines 170-172: I would recommend improving this sentence: ” From this perspective, many efforts were made to develop an efficient SE protocol for A. angustifolia and expand available methods its mass propagation and conservation.” - From this perspective, much effort has been made to develop an effective SE protocol for A. angustifolia and to expand the methods available for its mass propagation and conservation.

R: We changed the sentence, as suggested.

- lines 182-184: “The pre-maturation step triggers early SE polarization and individualization from PEM III in A. angustifolia. Removing the PGR from the culture medium and its supplementation with maltose, and PEG usually trigger this process”. - The pre-maturation step triggers early somatic embryo polarization and individualization from PEM III in A. angustifolia. This process is usually triggered by the removal of PGR from the culture medium and its replenishment with maltose and PEG. (or The pre-maturation step, when PGRs are replaced with maltose and PEG in the culture medium, induces polarization of early somatic embryos in A. angustifolia and their individualization from PEM III.)

R: We changed the sentence, as suggested.

- Lines 213 – 214: I do not understand this sentence: “Additionally, SE-specific genes, biochemical markers are important to understanding the regulatory mechanisms of SE.” Do you mean “In addition to SE-specific genes, biochemical markers are also important for understanding the regulatory mechanisms of SE”?

R: In fact, the meaning of the sentence was incorrect. We modified it as suggested.

- I would like to draw your attention also to some typing errors - in the abstract, line 13 - hystodifferentiation instead of histo...; line 96 - Araucariales e Cupressales instead of A. and C.?

R: We corrected the typos.

Reviewer 2 Report

The authors of the review “Somatic embryogenesis in conifers: one clade to rule them all?” used an integrated approach in order to cover the advances of knowledge related to somatic embryogenesis of conifers (Pinacea + non-Pinacea conifers) and Gnetales and discussed the state –of-the-art, shedding light on the similarities and current bottlenecks.

Authors have compared the number of reports since 2010 dealing with the somatic embryogenesis (SE) with Pinacea and non-Pinacea as a parameter to illustrate that there might be a preference for working with Pinacea species. I am not in agreement with authors to use this approach. There are seven conifer families within Coniferales (cone-bearing plants) representing 71 genera and roughly 620+ species (Table 1.2, Williams, Conifer Reproductive Biology 2009). If authors want to challenge themselves doing the comparison between Pinacea and non-Pinacea species, I would start with the proportion of different genera, species in different families, their economical importance as well as their geographical distribution. Then comparison of zygotic embryogenesis of these families should be presented as well as the size of their genome. According to me these are more important factors to be considered when comparing the interest in Pinacea and non-Pinacea somatic embryogenesis. Based on this comment, Table 1 (Table is even incomplete, missing several genera and contains several spelling errors in the authors names) and part of Figure 1 should be deleted from the manuscript.

Family

Genera

Cupressaceae sensu latob

Actinostrobus, Austrocedrus, Callitris, Calocedrus,

Chamaecyparis, X Cupressocyparis, Cupressus,

Diselma, Fitzroya, Fokienia, Juniperus, Libocedrus,

Microbiota, Neocallitropsis, Papuacedrus,

Platycladus, Pilgerodendron, Tetraclinis, Thuja,

Thujopsis, Widdringtonia, Xanthocyparis

Arthrotaxis, Cryptomeria, Cunninghamia, Glyptostrobus,

Metasequoia, Sequoia, Sequoiadendron, Taiwania,Taxodium

Pinaceae

Abies, Cathaya, Cedrus, Keteleeria, Larix, Nothotsuga,

Picea, Pinus, Pseudolarix, Pseudotsuga and Tsuga

Araucariaceae

Araucaria, Agathis, Wollemia

Podocarpaceae

Acmopyle, Afrocarpus, Dacrycarpus, Dacrydium,

Falcatifolium, Halocarpus, Lagarostrobos,

Lepidothamnus, Manoao, Microcachrys, Microstrobos,

Nageia, Parasitaxus, Phyllocladus, Podocarpus,

Prumnopitys, Retrophyllum, Saxegothaea,

Sundacarpus

Sciadopityaceae

Sciadopitys

Cephalotaxaceae

Cephalotaxus

Taxaceae

Amentotaxus, Austrotaxus, Pseudotaxus, Taxus, Torreya

Line 11 and 12

Authors have written that ….’SE is favoured over the methods of vegetative propagation due to the possibility to scale up the propagation using bioreactors and enable process such as’ …. If authors are making the statement in general for somatic embryogenesis, then I agree, however, the proper references are missing. If authors refer to somatic embryogenesis of conifers (title of the manuscript), then the sentence must be re-written. Cryo-preservation of embryogenic tissue is the biggest advantage of SE over other propagation techniques. Reports using coniferous species and bioreactors are still quite rare compared to Angiosperms.  

Line 79 …’ culturing the primary explant’… specification would be needed. In majority of cases the megagametophytes, isolated immature, or mature zygotic embryos were used as the primary explant.

Authors should consider that there is a very big diversity in the development of SE protocols even with Pinaceae. I would consider as a model species only Picea abies L. Karst. Most of Pinus species have less developed protocols and are still quite far away from the deployment stage.

I have an objection towards using the term callus for SE of conifers. From the initiation phase the embryogenic tissue is made up of 2 distinct cell types (embryogenic and suspensors) having completely different destinies during the development of somatic embryos. The whole manuscript must be screened for this serious discrepancy. Authors also wrongly use the term globular and torpedo somatic embryos with Araucaria. For conifer zygotic and somatic embryogenesis, the terminology early-precotyledonary, precotyledonary and cotyledonary embryos is used.

The part covering somatic embryogenesis of GnetaIes, Araucariales and Cupressales is comprehensively written. I would recommend authors to re-write the manuscript as a review focused on SE of other coniferous species not including Pinaceae.    

Spelling mistakes, forgotten brackets

Author Response

- Authors have compared the number of reports since 2010 dealing with the somatic embryogenesis (SE) with Pinacea and non-Pinacea as a parameter to illustrate that there might be a preference for working with Pinacea species. I am not in agreement with the authors to use this approach. There are seven conifer families within Coniferales (cone-bearing plants) representing 71 genera and roughly 620+ species (Table 1.2, Williams, Conifer Reproductive Biology 2009). If authors want to challenge themselves doing the comparison between Pinacea and non-Pinacea species, I would start with the proportion of different genera, species in different families, their economic importance as well as their geographical distribution. Then the comparison of the zygotic embryogenesis of these families should be presented as well as the size of their genome. According to me, these are more important factors to be considered when comparing the interest in Pinacea and non-Pinacea somatic embryogenesis. Based on this comment, Table 1 (The table is even incomplete, missing several genera, and contains several spelling errors in the author's names) and part of Figure 1 should be deleted from the manuscript.

R: We agree with the reviewer's excellent remarks and have reassessed our entire approach to the manuscript. We sought to improve consistency and up-to-dateness in gymnosperm phylogeny data, included topics suggested by the reviewer (Taxonomical considerations and Zygotic Embryogenesis in Conifers), omitted Table 1 and Figures 1 and 2, and revised all other information included in the MS. We genuinely hope we have answered the queries raised here and thank you for the solid and thorough analysis of our MS.

- Lines 11 and 12

Authors have written that ….’SE is favored over the methods of vegetative propagation due to the possibility to scale up the propagation using bioreactors and enable process such as’ …. If authors are making the statement in general for somatic embryogenesis, then I agree, however, the proper references are missing. If the authors refer to the somatic embryogenesis of conifers (title of the manuscript), then the sentence must be rewritten. Cryo-preservation of embryogenic tissue is the biggest advantage of SE over other propagation techniques. Reports using coniferous species and bioreactors are still quite rare compared to Angiosperms.

R: We slightly modified the quoted sentence and included the reference Egertsdotter et al. (2019), aiming to give greater support to the statement. We agree that, compared to angiosperms, the use of bioreactors in conifers is not yet extensively used. However, as presented in the review included as a citation, this application has been expanded in recent years, and we would like to highlight this advance.

Reference: Egertsdotter, U., Ahmad, I., & Clapham, D. (2019). Automation and scale-up of somatic embryogenesis for commercial plant production, with emphasis on conifers. Frontiers in Plant Science, 10, 109.

- Line 79 …’ culturing the primary explant’… specification would be needed. In the majority of cases the megagametophytes, isolated immature, or mature zygotic embryos were used as the primary explant.

R: We included the specification, as suggested.

- Authors should consider that there is a very big diversity in the development of SE protocols even with Pinaceae. I would consider as a model species only Picea abies L. Karst. Most  Pinus species have less developed protocols and are still quite far away from the deployment stage.

R: We agree with the reviewer and have modified the entire MS as suggested.

- I have an objection to using the term callus for SE of conifers. From the initiation phase, the embryogenic tissue is made up of 2 distinct cell types (embryogenic and suspensors) having completely different destinies during the development of somatic embryos. The whole manuscript must be screened for this serious discrepancy. Authors also wrongly use the term globular and torpedo somatic embryos with Araucaria. For conifer, zygotic and somatic embryogenesis, the terminology early-precotyledonary, pre-cotyledonary and cotyledonary embryos are used.

R: We agreed with the reviewer and revised the entire manuscript, and corrected it as suggested. The term "callus" only remained in cases where the article indicates that there was no somatic embryo formation or when it was impossible to identify this information in the original article. Regarding the terminologies referring to the stages of development of the embryos of A. angustifolia, we agreed with the reviewer and made the modifications, as suggested.

- The part covering somatic embryogenesis of GnetaIes, Araucariales, and Cupressales is comprehensively written. I would recommend authors to re-write the manuscript as a review focused on SE of other coniferous species, not including Pinaceae.

R: Our approach in drawing a brief parallel with SE protocols for species of the family Pinaceae was precisely to broaden the level of discussion and bring greater support to the significant gap that exists in SE works for non-Pinaceae conifers. Our intention was never to write a review approaching these two large clades homogeneously, precisely because we understand that it would be impossible to maintain the same level of detail and build a consistent manuscript. We still believe it is vital to keep the mentions made regarding SE work in Pinaceae, to serve as a background for the central point of our review, which is to delve deeper into SE in non-Pinaceae. We have extensively modified the MS and hope that this issue is addressed.

Reviewer 3 Report

This ms has very interesting topic, but unfortunately it does not fullfil the expectations that reader has after seeing the abstract. The ms is not a true comparison of SE in Pinaceae and non-Pinaceae, and it does not really answer to the questions raised in the abstract. The idea behind the ms is good, but the realization fails.

The ms is written in careless way: the contents of tables, figures and text is not consistent with each other, and indeed very many references, especially from the Tables are missing in the reference list. Just to give a few examples on inconsistencies: 

- in line 88 authors state that there are 48 reports of SE in Pinaceae, but in their own Figure 1  they mention 48+23+10.. plus all the ones e.g. on Picea that are completely missing form the ms

- in Fig 1 Cupressus belongs to Araucaliales, while in Table 2 it belongs to Cupressales...

- SE report in Gnetum is in Table 2, but missing in Figure 1

- Bornman 1975 discussed in the text is missing in Table 2

- in text plant regeneration was achieved in E, foliata, while in Table 2 only embryogenic callus was produced

- in E. gerardiana test tells about obtaining somatic embryos, but in Table 2 the same study produced embryogenic allus and shoot buds

etc etc

Both the tables need more concise layout to be more easy to read.

The term "callus" is used misleadingly and with several meanings.. By definition, callus is undifferentiated tissue, and it cannot really thus be embryogenic i.e. contain proembryos... Please check and correct throughout the ms if callus is callus (= non-embryogenic) or not callus but embryogenic tissue / culture...

Discussion on gene expression in Araucaria would provide a possibility to actually compare it with gene expression observed in Pinaceae SE, but this comparison is missing in the ms.

The non-Pinaceae genera Sequia, Thuja, Torreya, Cunninghamia and Chamaecyparis are shown in Table and Figures, but not discussed at all in the text although in many of them there are reports on successfull SE.

Conclusions made for common points and dis-similarities in non-Pinaceae SE are also mostly true for Pinaceae SE, so the overall value of this review, taking also into account the numerous inconsistencies remains not so high. 

No special comments on language

Author Response

- This ms has a very interesting topic, but unfortunately, it does not fulfill the expectations that the reader has after seeing the abstract. The ms is not a true comparison of SE in Pinaceae and non-Pinaceae, and it does not answer the questions raised in the abstract. The idea behind the ms is good, but the realization fails.

R: We appreciate your careful revision and accept all the criticisms related to the previous format of the manuscript. We worked hard to improve the consistency of the manuscript and thus meet the reader's expectations. This new version includes several improvements based on the excellent feedback we received from all reviewers.

- The ms is written carelessly: the contents of tables, figures, and text are not consistent with each other, and indeed very many references, especially from the Tables are missing from the reference list.

R: We apologize for the earlier inconsistencies found in the manuscript. We double-checked all the information contained in the text, tables, and figures to improve the consistency of the information in the review.

- in line 88 authors state that there are 48 reports of SE in Pinaceae, but in their Figure 1  they mention 48+23+10.. plus all the ones e.g. on Picea that are completely missing from the ms

R: We omitted this figure, as suggested by reviewer 2. Anyway, we have revised and corrected all inconsistencies in the MS.

- in Fig 1 Cupressus belongs to Araucaliales, while in Table 2 it belongs to Cupressales...

R: We omitted this figure, as suggested by reviewer 2.

- SE report in Gnetum is in Table 2, but missing in Figure 1

R: We omitted this figure, as suggested by reviewer 2.

- Bornman 1975 discussed in the text is missing in Table 2

R: We included it in Table 1.

- in text plant regeneration was achieved in E, foliata, while in Table 2 only embryogenic callus was produced

R: We corrected this information.

- in E. gerardiana test tells about obtaining somatic embryos, but in Table 2 the same study produced embryogenic callus and shoot buds

R: We corrected this information.

Both the tables need a more concise layout to be easier to read.

R: We have revised and reformatted the tables as suggested.

- The term "callus" is used misleadingly and with several meanings. By definition, callus is undifferentiated tissue, and it cannot thus be embryogenic i.e. contain proembryos... Please check and correct throughout the ms if the callus is callus (= non-embryogenic) or not callus but embryogenic tissue/culture...

R: We agreed with the reviewer and revised the entire manuscript, and corrected it as suggested. The term "callus" only remained in cases where the article indicates that there was no somatic embryo formation or when it was impossible to identify this information in the original article.

- Discussion on gene expression in Araucaria would provide a possibility to compare it with gene expression observed in Pinaceae SE, but this comparison is missing in the ms.

R: We have included the following paragraph in the discussion: "Studies involving SE-related gene expression in Pinaceae species have also been reported. Vestman et al. (2011) reported that a member of the ARGONAUTE family from white spruce is required for proper shoot and root meristem differentiation during embryo development. Important genes for normal embryo patterning, such as putative homologs of SERK and WOX2, were also identified as being expressed both during the proliferation and differentiation of early embryos and late embryo development (Vestman et al., 2011). Rupps et al. (2016), investigating SE in Larix decidua, also reported transcript accumulation of LdLEC1 and LdWOX2 during early embryogenesis, whereas LdSERK reveal increased expression during later development. In this context, the expression control of the mentioned genes seems to be conserved during the SE of Pinaceae and non-Pinaceae species."

- The non-Pinaceae genera Sequoia, Thuja, Torreya, Cunninghamia, and Chamaecyparis are shown in Table and Figures, but not discussed at all in the text although in many of them, there are reports on successful SE.

R: We included those reports in the text, as suggested.

- Conclusions made for common points and dis-similarities in non-Pinaceae SE are also mostly true for Pinaceae SE, so the overall value of this review, taking also into account the numerous inconsistencies remains not so high. 

R: The common points and dissimilarities presented in the conclusion are indeed similar to what was observed for the Pinaceae species. However, in our opinion, this is valid and pertinent information, since our approach in the review was precisely to evaluate the most sensitive points for optimizing culture conditions during somatic embryogenesis of non-Pinaceae species. Furthermore, we would like to point out that the entire manuscript was revised to improve its consistency and cohesion. Therefore, we believe that the current version of the manuscript can meet the quality standards.

Round 2

Reviewer 2 Report

Authors significantly improved the quality of the manuscript, considered all suggestions and comments. I recommend the manuscript for publishing it. A published review will be an excellent source of information for the scientific community. 

Reviewer 3 Report

The ms has improved a lot from its previous version. The authors have taken most of my comments into account and revised the ms accordingly.

I still found the questionable term "embryogenic calli" on r. 216 - please change it to embryogenic cultures.

English is mostly ok, please change the embryogenic calli to embryogenic cultures.